

# A novel approach to assessing the bioavailability of biopeptide inhibitor of HMG CoA reductase from germinated and ungerminated Kara Kratok (*Phaseolus lunatus* L.)

Cahyo Budiyanto, Andriati Ningrum, Agnes Murdiati and Retno Indrati

Faculty of Agricultural Technology, Universitas Gadjah Mada, Department of Food and Agricultural Product Technology, Yogyakarta, Yogyakarta, Indonesia

## ABSTRACT

**Background**. The bioavailability of biopeptide compounds is a development challenge, mainly because of their resistance to the digestion system. This study aimed to determine the bioavailability of HMG CoA reductase biopeptide inhibitors from germinated and ungerminated Kara Kratok (*Phaseolus lunatus* L.).

**Methods**. Germinated and ungerminated brown *P. lunatus* were simulated for digestion enzyme *in vitro* (120 minutes for pepsin and pancreatin), followed by an *in situ* method for absorption. Perfusate samples were measured for the absorption percentage, inhibition of HMG CoA reductase, molecular weight (MW), peptide concentration, and hydrolysis degree (%DH).

**Results**. The results showed that germinated brown *P. lunatus* exhibited the highest absorption (32.42%), and the percentage of HMG CoA reductase inhibition during enzymatic digestion was at 210 minutes (87.51%), with MW < 10 kDa, peptide concentration of 2.39 mg/mL, and %DH of 48.90%. These findings suggest that germinated brown *P. lunatus* is a potent HMG CoA reductase inhibitor with significantly higher bioavailability than that of its ungerminated counterpart. This finding underscores its superiority in this context and open new possibilities for biopeptide research.

## INTRODUCTION

The bioactivity of biopeptides, during digestion, at a specified target is crucial. As suggested by *Bhandari et al. (2020)*, the differences in bioactivity between various biopeptide compounds indicate their bioavailability, absorption, and susceptibility to breakdown into inactive fragments by physiological digestive enzymes. In this complex process, peptide transporters in the intestine play a key role, facilitating the passage of smaller peptides through digestive enzymes, while oligopeptides are transported passively through hydrophobic areas of epithelial membranes. This intricate process underscores the complexity of our research and the importance of understanding the role of peptide transporters in the absorption of biopeptides.

Corresponding author
Retno Indrati, indrati@ugm.ac.id

The bioavailability of macronutrients, carbohydrates, proteins, and fats is usually high, with more than 90% of the amount ingested being absorbed and utilized in the human body (*Schönfeldt, Pretorius & Hall, 2016*). The bioaccessibility and bioavailability of several biopeptide compounds increase after digestive enzymes hydrolyze them. Therefore, process technology, such as encapsulation, is needed. This method protects the active biopeptide compounds after they enter the body, the process of releasing and degrading digestive enzymes (*Indrati, 2021*). In the gastrointestinal tract, biopeptides pass the intestinal epithelial barrier before arriving at the organ's target and act its bioactivity (*Amigo & Hernández-Ledesma, 2020*).

Peptide absorption occurs in two processes, membrane hydrolysis and intracellular hydrolysis. In membrane hydrolysis, peptides with a high affinity for the peptidase found in the brush border membrane hydrolyze into shorter peptides. In contrast, peptides resistant to the peptidase transferred intact through the small intestinal epithelial cells and then hydrolyzed by the peptidase in the cytoplasm, which is referred to as intracellular hydrolysis (*Xu et al., 2019*). According to *Miner-Williams, Stevens & Moughan (2014)*, bioactive peptide compounds can generally reach the target organ intact because they are resistant to both the processes of membrane hydrolysis and intracellular hydrolysis.

Research conducted by *Aiello et al. (2018)* on soybean bioactive peptide compounds IAVPTGVA, LPYP, and IAVPGEVA as HMG CoA reductase inhibitors against Caco-2 cells showed the inefficiency of bioactive peptide compounds during transport in the intestine due to high hydrolysis by brush border enzyme. *Marques et al. (2018)* stated that bioactive peptide compounds with a molecular weight of $\leq$ 3 kDa from cowpea protein hydrolysate against Caco-2 cells showed that the absorption of the MELNAVSVVHS peptide passed through the cell monolayer membrane. No studies have reported on the bioavailability of HMG CoA reductase inhibitor from biopeptide compound, nor on statin compound which is maximum 30% with absorption percentage maximum 98% (*Neuvonen, Backman & Niemi, 2008*). This study aimed to investigate the bioactivity of biopeptide HMG CoA reductase inhibitor from *P. lunatus* during digestion *via in vitro* method and its bioavailability using the *in situ* method. *P.lunatus* has the potential as a biopeptide source to inhibit the HMG CoA reductase. This is because it is rich in leucine which is around 8.84–9.84 g/100 g protein (*Palupi et al., 2022*; *Seidu, Osundahunsi & Osamudiamen, 2018*), although this legume has low commercial value in addition to its low utilization in Indonesia. HMG CoA reductase is an enzyme that plays an important role in the mevalonate pathway for HMG-CoA conversion to cholesterol (*Murphy et al., 2020*). This enzyme is inhibited by statins which compete with HMG-CoA because it has the same hydrophobicity, rigidity, and covalently bonded to HMG-like sites (*Istvan, 2003*).

## MATERIALS AND METHODS

### *P. lunatus* beans and chemicals

*P. lunatus* beans were obtained from the Bondowoso local market, East Java, while white *Sprague Dawley* rats from PAU Pangan dan Gizi (Food and Nutrition Development Research Center) at Universitas Gadjah Mada (UGM) Yogyakarta. Chemical agents and

kits that were used in this study, include o-Phthaldialdehyde (OPA) from Sigma Aldrich for peptide concentration measurement and HMG CoA reductase kit (CS-1090 Merck) for assessing HMG CoA reductase inhibition.

### Preparation of germinated and ungerminated *P. lunatus*

Germinated and ungerminated brown *P. lunatus* samples were prepared according to the method introduced by *Chel-guerrero et al. (2012)* with some modifications. Clean *P. lunatus* seeds were soaked for 24 h in distilled water, continued in 0.2% (w/v) of sodium hypochlorite solution for 1 minute, and finally, they were cleaned with distilled water for three times. Ungerminated beans were obtained after this step, while germinated bean were obtained from 72 h of germination in a 16–18 °C dark room. The germinated *P. lunatus* beans were dried in cabinet dryer for 8 hours at 50–55 °C, and then processed into flour. Both samples were simulated *in vitro* for gastrointestinal digestion and continued with the *in situ* absorption method.

### Gastrointestinal digestion simulation

The simulation of gastrointestinal digestion was carried out following the methods described by *Minekus et al. (2014)* and *Moreno et al. (2020)* employing and *in vitro* simulation using pepsin and pancreatin. Samples of germinated and ungerminated *P. lunatus* were carefully simulated with pepsin and continued with pancreatin enzyme. The total reaction time for each sample was 240 min, with each enzyme taking 120 min and sampling every 30 min. The 30-minute sampling analyzed the peptide concentration, hydrolysis degree (%DH), molecular weight (MW), and the percentage inhibition of HMG CoA reductase.

### Ethics approval

Ethics approval for protocols and procedures for the experiments was granted by the Lembaga Penelitian dan Pengujian Terpadu—Integrated Laboratory for Research and Testing—(LPPT) of Universitas Gadjah Mada (no. 00073/04/LPPT/I/2024).

### Animals experiment—*In situ* absorption

The *in situ* absorption was performed following *Xiao et al. (2024)*. Sample size calculation with quantitative data in the endpoint followed *Charan & Kantharia (2013)*, using two groups of healthy male *Sprague Dawley* (SD) rats obtained from the PAU Pangan dan Gizi (Food and Nutrition Development Research Center) of UGM Yogyakarta. The body weight of the rats ranged from 236 to 315 grams ($n = 3$). A total of six SD rats were used to minimize animal usage. Each SD rat was kept in a cage measuring 20 cm × 20 cm × 24 cm, with an environment maintained at 22.5 °C and RH of 50%. The animals were subjected to a 12-hours light-dark cycle and fed with a standard diet for rodent. They had access to reverse osmosis (RO) water ad libitum prior to 24 h fasting period with a normal water intake. Each SD with normal physiological functions such as normal respiratory function, was then anesthetized by injection into the thigh with ketamine 60 mg/kg bodyweight following the method of follows the method of *Navarro et al. (2021)*. After the rat fainted, indicated by its immobility, dilating pupils, and cessation of breathing, the surgical procedure was then

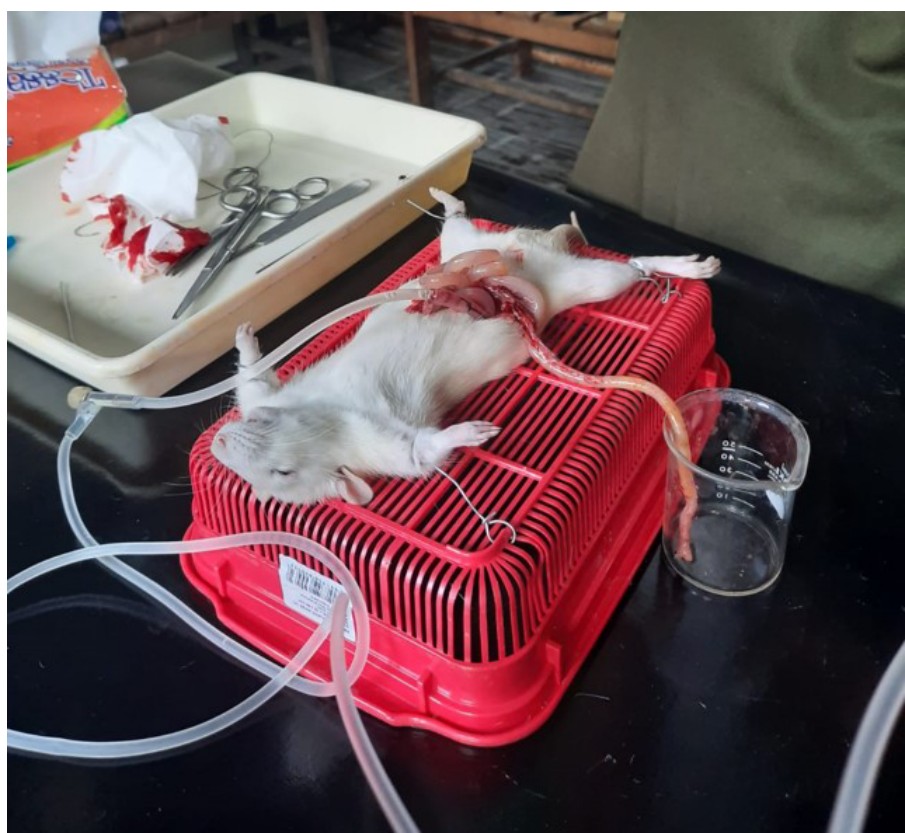

**Figure 1** **Final preparation on *Sprague Dawley* for absorption test with *in situ* method.** $n = 3$ animals for each sample.

carried out. A midline longitudinal incision was made to open the abdomen. Then plastic tubing (4 mm o.d.) was used for cannulation at 10 cm from the jejunum and 10 cm before the cecum for sample collection, followed by perfusion with 0.9% sodium chloride solution with a dosage of 0.2 ml/minute for 120 min to flush the retaining food in the digestion tract. Samples were injected with the same flow and time and collected at 30, 60, 90, and 120 min (Fig. 1). Ungerminated *P. lunatus* beans were used as the control samples ($n = 3$), while the germinated ones as the treatment's samples ($n = 3$). The perfusate samples were analyzed for the percentage of absorption *via* peptide concentration, molecular weight, and the percentage inhibition of HMG CoA reductase. At the end of the experiment there was no living SD because all of them were euthanized using a lethal dose of ketamine 100 mg/kg body weight administered *via* intramuscular injection. Subsequently, the remains were incinerated at the PAU Pangan dan Gizi (Food and Nutrition Development Research Center) UGM outside the laboratory. For reporting our animal research, the complete ARRIVE Guidelines checklist was used which is included in the appendices.

The absorption percentage was calculated from the peptide content difference before and after the sample passed the digestion tract used. A higher value meant higher absorption.

**Table 1  Reaction mixture for inhibition of HMG CoA reductase activity.**

| Sample | 1x Assay Buffer (µl) | Pravastatin (µl) | NADPH (µl) | HMG-CoA (µl) | HMG CoA reductase (µl) | *P. lunatus* (µl) |
|---|---|---|---|---|---|---|
| Blank | 184 | – | 4 | 12 | – | – |
| Negative control | 182 | – | 4 | 12 | 2 | – |
| Positive control | 181 | 1 | 4 | 12 | 2 | – |
| Samples (biopeptide candidates) | 178 | – | 4 | 12 | 2 | 10 |

## Peptide content and degree of hydrolysis

A peptide sample solution of 0.4 mL was added with three mL of OPA reagent and then incubated for 20 min at room temperature. Measurement of these samples used a spectrophotometer (GENESYS 10S) at absorbance λ 340 nm (*Agustia et al., 2023*). Standard curves were set by using tryptophan solution, and %DH was calculated using the equation:

$$\%DH = \frac{(Ntx - Nt0)}{(Ntot - Nt0)} \times 100\%$$

where Ntx is the sample hydrolysis at x-hour, Nt0 is the sample hydrolysis at 0-hour, and Ntot is the total quantity of the sample hydrolysis for 4 h at 105 °C with 10 mL HCl 6 M.

## Molecular weight

Molecular weight was analyzed by following the SDS-PAGE method (*Laemmli, 1970*) using 5% resolving gel and 13% separating gel. SDS sample buffer contained 0.5M Tris-HCl pH 6.8, 87% glycerol (w/v), 10% SDS (w/v), and 0.5% bromophenol blue (w/v). Distilled water was used to dilute the peptide extracts at a ratio of 1:2. The sample solution was firstly heated for 4 min at 100 °C, then loaded 20 µl samples or 5 µL of standard protein marker was loaded into each well. The gel ran at 220V for 60 min, continued with 30 min of soaking in distilled water, and repeated three times. The final step was staining with 0.2% Coomassie Brilliant Blue R-250 which contained 50% methanol, 10% acetic acid, and 40% distilled water.

## HMG CoA reductase inhibition

The percentage of the HMG CoA reductase inhibition was measured using the Sigma Aldrich CS-1090 kit and followed the method established by *Hermanto et al. (2020)* with the quantities of each reaction volume specified in Table 1. The analysis purpose of HMG CoA reductase inhibition during gastrointestinal enzymatic digestion was to see the changes of the peptide ability and its resistance to the digestion enzyme, while in the absorption was to know its bioavailability.

The 200 µL mixture samples and reagents were measured at 340 nm wavelength for 10 min reading. Enzymatic activity was calculated using the equation:

$$\text{Unit per mgP} = \frac{(\Delta\text{Absorbance 340 nm sampel} - \Delta\text{Absorbance 340nm blangko})}{12.44 \times V \times 0.6 \times LP} \times TV$$

where:

12.44 = During reaction need 2 NADPH (the NADPH coefficient at 340 nm is 6.22/mM cm)
TV = Total reaction volume (one mL)

V = Enzyme volume used

0.6 = Enzyme concentration in mg-protein (mgP)/mL

LP = Ligh path (one cm for cuvettes and 0.55 cm for plate)

The equation below is used to calculate the percentage of inhibition:

$$\% \ inhibition = \frac{\text{Enzyme Activity (non−inhibitor)} − \text{Enzyme Activity (pravastatin or sample)}}{\text{Enzyme Activity (non−inhibitor)}}$$
$$\times 100\%.$$

### Statistical analysis

The quantitative statistical analysis of the data, showing significant differences ($P < 0.05$) between treatments, was conducted using SPSS IBM 23 software. A oneway analysis of variance (ANOVA) was used for this statistical analysis. If a significant difference was observed, the analysis continued with Duncans' Multiple Range Tests (DMRT). Total experiment's data was nine that obtained from three batch samples and three times analysis of each batch.

## RESULTS

### Peptide content and degree of hydrolysis

The results showed an incremental degree of hydrolysis (Fig. 2A) and peptide concentration (Fig. 2B) for both germinated and ungerminated *P. lunatus* until 210 min and then decreased at the end of the digestion (240 min). Statistically, for every 30 min of digestion, a significance difference ($P < 0.05$) was observed between the germinated and ungerminated samples. In addition, germinated *P. lunatus* showed a higher degree of hydrolysis and peptide concentration than its ungerminated counterpart.

### Molecular weight

The digestion enzymes of germinated *P. lunatus* were eight fractions higher than those of the ungerminated ones. The molecular weight of germinated *P. lunatus* was in the range of 6.9–38.7 kDa (Fig. 3A), while that of the ungerminated *P. lunatus* was in the range of 13.5–43.2 kDa (Fig. 3B). In both samples of the germinated and ungerminated *P. lunatus*, the molecular weight's band showed increased thickness over time than their initial form. Meanwhile, the aim of the molecular weight analysis in the perfusate of *in situ* simulation was to assess the unabsorbed peptide in the digestion system, which showed that the germinated *P. lunatus* had a thinner band than the ungerminated ones (Fig. 3C). Both samples showed the same molecular weight of the unabsorbed peptide, which was 15 kDa and 75 kDa.

### Inhibition of HMG CoA reductase

The percentage inhibition of HMG CoA reductase from germinated *P. lunatus* was found to be higher than and significantly different ($P < 0.05$) from that of the ungerminated beans during enzymatic digestion time (Fig. 4A). Both germinated and ungerminated *P. lunatus* hydrolysate showed incremental HMG CoA reductase inhibition during digestion time

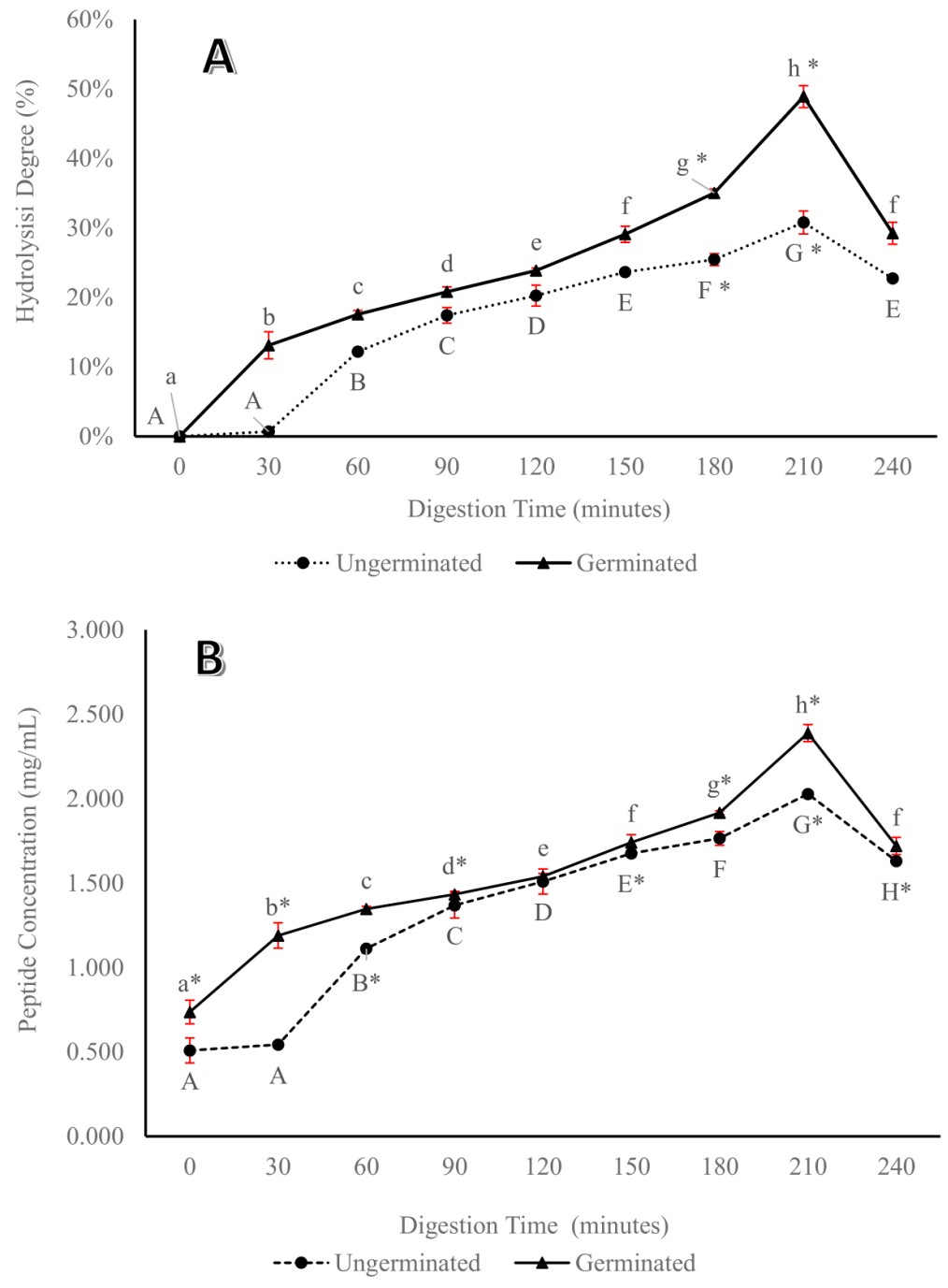

**Figure 2** **Percentage of hydrolysis degree (A) and peptide concentration (B) from germinated and ungerminated *P. lunatus* during digestion gastrointestinal simulation *via in vitro* method.** Within each graph, lines with different letters indicate significant differences (Duncan's Multiple Range Test, $P < 0.05$). An asterisk (*) denotes a significant difference across all data (Duncan's Multiple Range Test, $P < 0.05$).

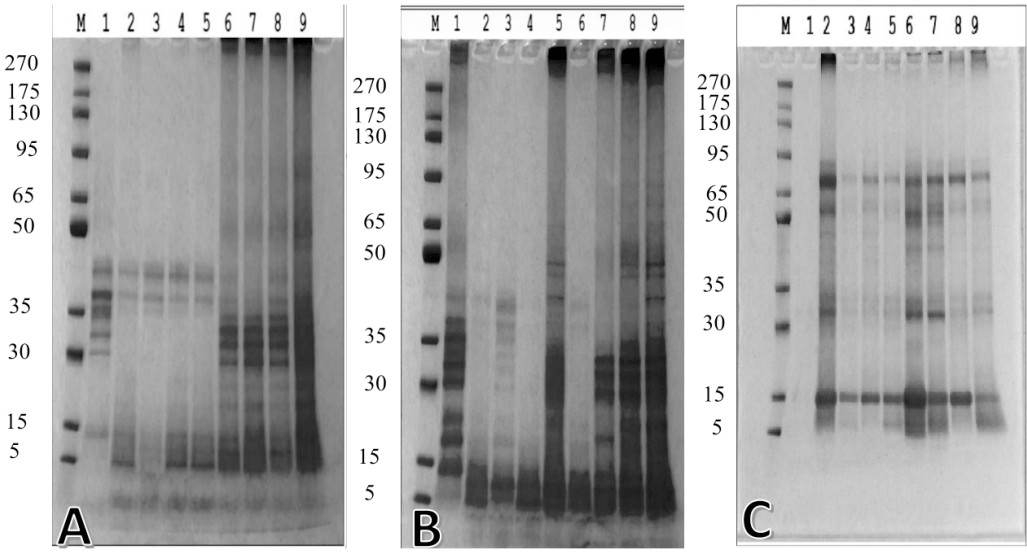

**Figure 3** The molecular weight of (A) ungerminated and (B) germinated *P. lunatus* during gastrointestinal digestion simulation *via in vitro* method—lane M (marker), lane 1–9 (digestion time 0, 30, 60, 90, 120, 150, 180, 210, and 240 min), (C) biopeptide unabsorbed during *in situ* method at 30, 60, 90, and 120 min—lane M (marker), lane 2–5 (ungerminated *P. lunatus*), lane 6–9 (germinated *P. lunatus*).

with a maximum inhibition at 210 min. Hydrolysate of germinated *P. lunatus* by pepsin and pancreatin also gave a higher percentage of inhibition and was significantly different ($P < 0.05$) compared to pravastatin as a control, but not for the ungerminated ones because higher percentage of HMG CoA reductase inhibition from pravastatin only started at 180 min. This pattern is in line with the peptide concentration pattern and the formation of low molecular weight proteins results at the end of enzymatic digestion.

Perfusate obtained during *in situ* absorption was also analyzed for its ability to inhibit HMG CoA reductase, and the results showed that the peptides were not absorbed in the digestive tract. The ability to inhibit HMG CoA reductase of peptide in perfusate at the end digestion time from germinated *P. lunatus* was lower than that of the ungerminated ones. Statistically, they were significantly different ($P < 0.05$) from each other (Fig. 4B). Figure 4B shows a reduction of its ability as HMG CoA reductase inhibitor from the beginning of the absorption time to 120 min of absorption.

### Percentage of peptide absorption *via in situ* method

The highest concentration of peptides absorbed was in the germinated sample, which was significantly different ($P < 0.05$) from that of the ungerminated one (Fig. 5).

## DISCUSSION

### Peptide content and degree of hydrolysis

The incremental hydrolysis degree of both germinated and ungerminated *P. lunatus* during enzymatic digestion is related to the composition and structure of phaseolin which is known for its susceptibility to enzymatic hydrolysis (*Montoya et al., 2008*). Pepsin

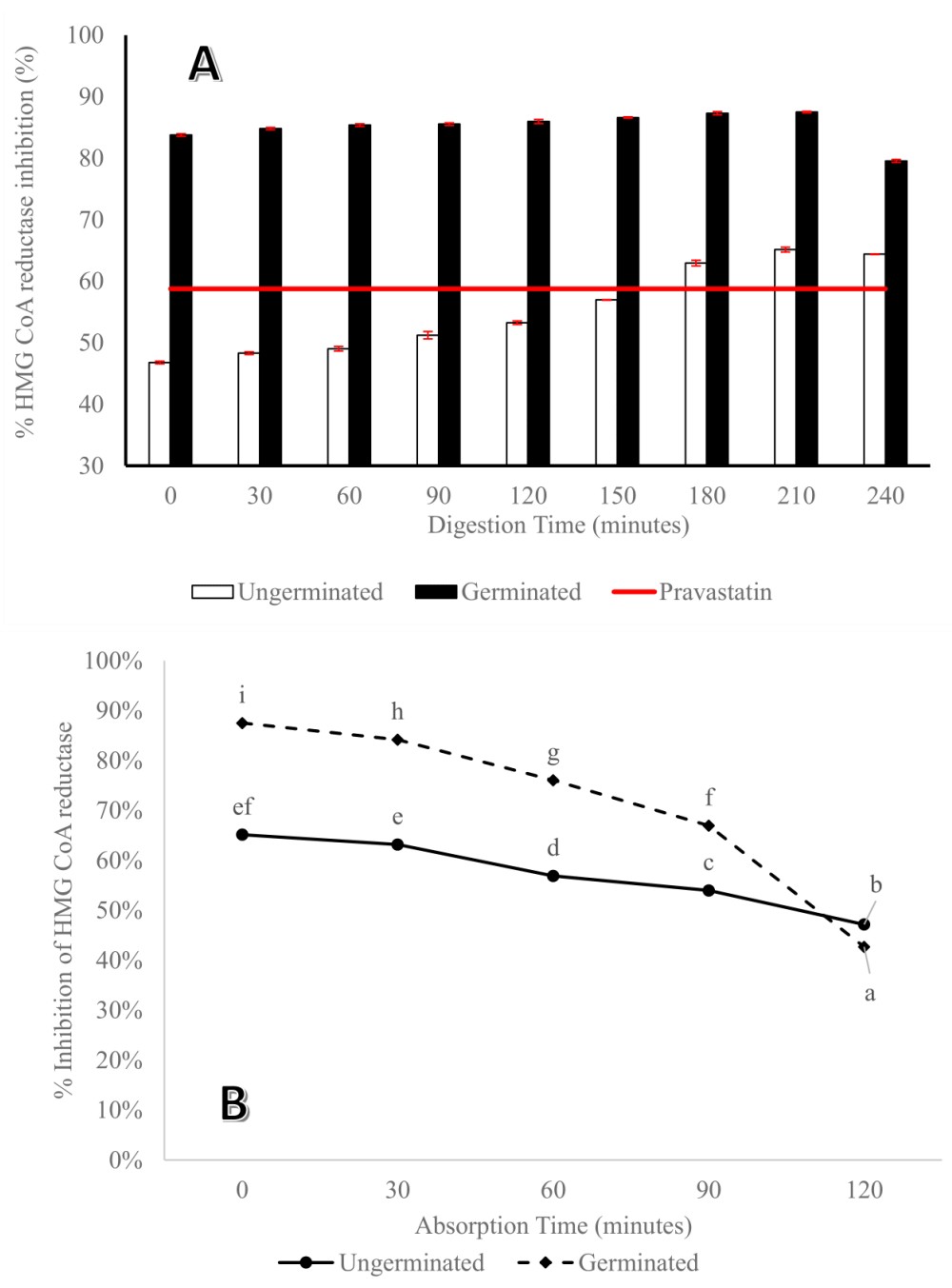

**Figure 4 Percentage of HMG CoA reductase inhibition from germinated and ungerminated *P. lunatus versus* pravastatin as a drug reference (A) during digestion gastrointestinal simulation *via* the *in vitro* method.** Within each graph, lines with different letters indicate significant differences (Duncan's Multiple Range Test, $P < 0.05$). An asterisk (*) denotes a significant difference across all data (Duncan's Multiple Range Test, $P < 0.05$). (B) In perfusate that contains unabsorbed peptide. Lines with different letters and an asterisk (*) showed a significant difference across all data (Duncan's Multiple Range Test, $P < 0.05$).

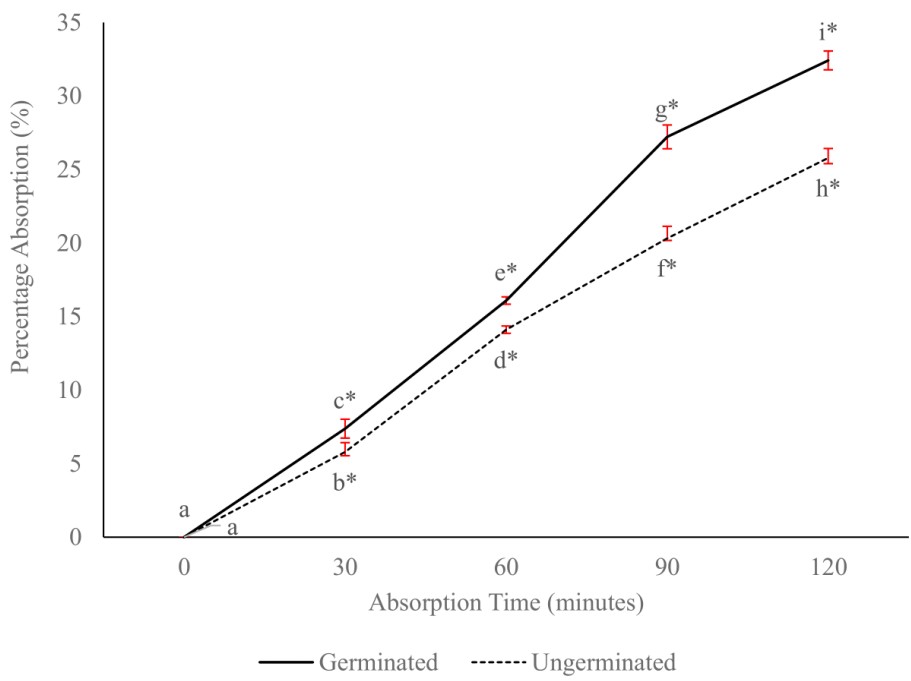

**Figure 5 Percentage absorption of germinated and ungerminated *P. lunatus* based on the calculation of peptide concentration before and after the *in situ* absorption method.** Lines with different letters and an asterisk (*) showed a significant difference across all data (Duncan's Multiple Range Test, $P < 0.05$).

and pancreatin hydrolysis have a specific preference location to cut the peptide. Pepsin cuts peptide bonds after phenylalanine, leucine, and glutamic acid, and rarely cuts after histidine and lysine even though it is close to leucine and phenylalanine (*Ahn et al., 2014*). Meanwhile, pancreatin hydrolyzes aromatic and branched-chain amino acids such as tyrosine, phenylalanine, tryptophan, alanine, and leucine, and cuts all amino acid residues except asparagine, glutamine, arginine, and lysine (*Andriamihaja et al., 2013*; *Ratnayani et al., 2019*). The maximum hydrolysis degree at 120 min of pepsin hydrolysis was 23.88% for germinated *P. lunatus* and 20.27% for the ungerminated beans. This finding is also reported by *Misquitta et al. (2023)* in legume hydrolysates using a combination of pepsin and pancreatin, which generally has %DH in the range of 20–46%.

The peptide concentration of germinated *P. lunatus* showed an increase starting from the 30 min of the pepsin hydrolysis phase, while the ungerminated samples started at the 60 min. This difference can be attributed to the storage protein of the germinated beans which has changed its shape from a folded to an unfolded structure. This starts by carbohydrase activity on cell wall damage, which continues with a protease that hydrolyzes the storage protein (*Halmer, Bewley & Thorpe, 1975*). *Ohanenye et al. (2021)* also found that the microstructure of the cell wall's softening and damage started after 24 to 48 h of germination. The incremental peptide concentration of germinated and ungerminated *P. lunatus* at the 210 min indicates the formation of a small peptide increment as enzymatic activity. Pepsin initiates the breakdown of proteins into smaller peptides in the stomach

that will be able to be hydrolyzed by pancreatin in the small intestine. In addition, pepsin breaks peptide bonds on certain sides that may be resistant to pancreatin hydrolysis (*Del Rio et al. 2021*; *Fu et al., 2021*). In other words pepsin hydrolysis in the early stages of digestion can increase the efficiency of protein hydrolysis by pancreatin enzymes in later stages.

Furthermore, germinated *P. lunatus* exhibited a higher degree of hydrolysis and peptide concentration than the ungerminated beans. This is related to a reduced amount of substrate or maximum protein hydrolysis products. These results are also aligned with those of *Sandoval-Sicairos et al. (2020)*. They studied the antioxidant activity of germinated and ungerminated *amaranth* during gastrointestinal digestion simulation. The results indicate the same pattern of increment until 210 min of digestion, followed by a decline at 240 min, which is aligns with the activity of pepsin and pancreatin that hydrolyze large protein molecules into small peptides.

## Molecular weight

The difference in the number of fractions formed between germinated and ungerminated *P. lunatus* is due to the effect of germination that softens and damages the microstructure of the *P. lunatus* seed cell wall, followed by the breakdown of stored protein by pepsin into smaller peptides, which is further hydrolyzed by pancreatin. The microstructure of the new cell wall softening and damage begins after 24 to 48 h of germination (*Ohanenye et al., 2021*). According to *Atudorei, Stroe & Codină (2021)*, the cell structure of the germinated seeds will undergo continuous changes as indicated by the separation of starch granules from protein, but the seeds still have a smooth surface due to increased hydrolytic enzyme activity. The thicker density band during 120 min of pepsin digestion provides a fairly high increase in short-chain peptides, especially in the germinated samples. According to *Luo et al. (2018)*, pepsin has highly active properties since the beginning of digestive hydrolysis. Most peptides (50 to 93%) are formed starting at 30 s of legume protein digestion. *Likittrakulwong, Poolprasert & Srikaeo (2021)* reported that during *in vitro* digestion, rice protein from paddy rice and germinated paddy rice was completely digested after pancreatin and no protein bands were detected. Lower MW was achieved in germinated *P. lunatus* than in ungerminated samples, this is related to the proteolysis reaction by proteases during germination (*Sandoval-Sicairos et al., 2020*). This finding is in line with that of *Yuan et al. (2019)* who reported a decrease in MW from 52.80 kDa to 14.67 kDa and 12.01 kDa during gastric and intestinal digestion in antioxidant biopeptides of tropical sea cucumbers (*Holothuria leucospilota*). *Bera et al. (2023)* also compared peptides from raw chickpeas and chickpea sprouts after going through the oral, gastric, duodenal, and brush border digestion phases. The results showed a doubling of free alpha-amino nitrogen after the seeds sprouted, which indicates an increase in amino acids and oligopeptides. However, there was a decrease in the number of larger peptides detected by MS/MS and a decrease in the number of peptides that were resistant to all four phases of digestion.

The absorption process is influenced by molecular weight. *Patel & Mirsa (2011)* stated that absorption will generally decrease exponentially at molecular weights above 300 Da. This is in line with our study in which the molecular weight of the unabsorbed peptide

was 15 kDa and 75 kDa as shown by the thickest band in Fig. 3C. Perfusate of germinated *P. lunatus* shows a thinner band than its ungerminated counterpart due to the formation of small peptides and amino acids as the proteolytic actions of endopeptidase and exopeptidase (*Smith & Morton, 2010*). According to *Newstead et al. (2011)*, peptides that can be bound by the PepT1 transporter in peptide absorption are limited to di- and tripeptides. However, it is difficult for long-chain peptides with peptides more than tetrapeptides because their binding capacity is only on molecules of approximately $13 \times 12 \times 11$ Å. *Liu et al. (2024)* stated that the dipeptide and tetrapeptide with hydrophobic and basic amino acids at the C-terminal can pass the intestinal epithelium. *Kiela & Ghishan (2016)* stated that dipeptides and tripeptides are absorbed with high efficiency in the small intestine, and 90% of the absorbed food protein is in the form of amino acids and 10% as dipeptides and tripeptides. *Xu et al. (2019)* reported that the bioavailability of absorbed peptides is influenced by various peptide properties such as size, molecular weight, amino acid sequence, and peptide resistance to enzymatic degradation.

## Inhibition of HMG CoA reductase

The percentage of the HMG CoA reductase inhibition from germinated and ungerminated *P. lunatus* at the end of enzymatic digestion was higher than that of pravastatin as the reference. This results is in line with that of *Hermanto et al. (2020)*, who reported that soy protein isolate using 0.2% papain enzyme at an incubation temperature of 50 °C for 3 h resulted an inhibition value of 95.65%. Shami tree pod extract (*Prosopis cineraria* L.) provides HMG CoA reductase inhibition of up to 78.1% (*Jaipal et al., 2022*). In legumes, peptides <3 kDa from cowpeas provide a percentage of HMG CoA reductase inhibition of 95.0%, which is higher than and significantly different from pravastatin (*Silva et al., 2018*). The ungerminated *P. lunatus* showed the lowest HMG CoA reductase inhibition until the end of pepsin digestion for 120 min (53.26%). This is in line with the findings of *Pak et al. (2006)* who hydrolyzed soybeans using pepsin with 45% inhibition. *Lammi, Zanoni & Arnoldi (2015)* also reported 17% inhibition of HMG CoA reductase at a maximum dose of 2.5 mg/ml biopeptide from lupin hydrolysate using pepsin enzyme, but higher inhibition (57%) if using trypsin enzyme at the same concentration. According to *Smith & Morton (2010)*, pancreatic enzymes contain trypsin, chymotrypsin, elastase, and row carboxypeptidases (A and B) that will degrade proteins into small peptides in large quantities and then bond to the HMG CoA reductase enzyme.

During absorption, dipeptides and tripeptides are absorbed in high efficiency in the small intestine, and 90% of the absorbed food protein is in the form of amino acids and 10% as dipeptides and tripeptides (*Kiela & Ghishan, 2016*). According to *Fan et al. (2018)*, peptides are composed fewer than six amino acids can pass through enterocytes with decreased absorption efficiency due to increasing chain length. In their review, *Liu et al. (2024)* stated that the dipeptide and tetrapeptide with hydrophobic and basic amino acids at the C-terminal can pass the intestinal epithelium. Perfusate of germinated *P. lunatus* showed lower inhibition of HMG CoA reductase than the ungerminated *P. lunatus*. This suggest that germination helps provide some small peptides and digestion enzymes continue the degradation into amino acids, dipeptide and tripeptide.

## Percentage peptide absorption *via in situ* method

The absorption process is influenced by the molecular weight and size of the peptide compound in the sample; *Mehrotra et al. (2024)* stated that absorption will generally decrease exponentially at molecular weights above 300 Da. *Keller (2013)* noted that small peptides with a composition of two or three amino acids can be absorbed directly through the apical membrane of enterocytes lining the small intestine *via* peptide transporters stimulated by PEPT-1 protons. Then, these small peptides will be broken down intracellularly into single amino acids. These results are also in line with the peptide concentration and MW of germinated and ungerminated *P. lunatus* after *in vitro* digestion simulation which showed that the concentration of peptides from the germinated samples was 2.388 mg/mL while that ungerminated samples was 2.028 mg/mL. The lowest MW fraction of the germinated *P. lunatus* was 6.9 kDa while that of the ungerminated samples was 13.5 kDa. Bioactive peptide compounds have significantly low transcellular permeability, which complicates their absorption into the portal vein through this transcellular pathway. On the other hand, the paracellular route requires very small molecules of less than 500 Da because absorption will go through TJ pores which have sizes ranging between 3 and 10 A° and are full of water. Molecules larger than 500 Da are generally unable to move through these small pores, although TJ can experience increased permeation, which makes the pores larger with a width remaining less than 20 nm (*Anderson & Van Itallie, 1995*; *Suzuki, 2013*). This percentage of absorption is still lower than the percentage reported in the research by *Pertiwi, Marsono & Indrati (2020)* on raw tempeh and cooked tempeh using the reverse intestinal absorption method. The highest absorption percentage was observed in the jejunum, which was 42.97% in raw tempeh samples and 49.43% in cooked tempeh samples. *Indrati (2021)* stated that there are four important steps in the effective absorption of bioactive compounds: (a) release from the food matrix, (b) incorporation into bile salt micelles, (c) absorption by epithelial cells, and (d) incorporation into chylomicron secretion into the lymphatic system. *Sepúlveda & Smith (1978)*, *Sepúlveda & Smith (1979)* stated that the smaller molecular weight of hydrophilic and neutral amino acids will be absorbed more slowly than hydrophobic amino acids with a larger molecular weight, while basic amino acids are absorbed at a moderate rate.

## CONCLUSIONS

The biopeptide of germinated *P. lunatus* exhibited higher bioavailability as an HMG CoA reductase inhibitor and absorption than the ungerminated beans. This observation is supported by the incremental peptide concentration and degree of hydrolysis peaking at 210 min during gastrointestinal simulation, as well as the high formation of lower molecular weight at the end of enzymatic hydrolysis. These promising findings not only encourage further research in this area but also provide a new avenue for the development of more effective biopeptide inhibitors, potentially revolutionizing the field of biochemistry and pharmaceuticals. The study is limited by *in vitro* and *in situ* method. Therefore, further research is needed for human *in vivo*.

## ACKNOWLEDGEMENTS

The authors would like to thank the Faculty of Agricultural Technology and the Lembaga Penelitian dan Pengujian Terpadu—Integrated Laboratory for Research and Testing—(LPPT) and the PAU Pangan dan Gizi (Food and Nutrition Development Research Center) for facilitating this research.

### Funding

Financial support for this study was provided by the Ministry of Education, Culture, Research, and Technology through a postgraduate research scholarship administered by the Directorate of Research, Universitas Gadjah Mada (grant numbers 048/E5/PG.02.00.PL/2024; 2781/UN1/DITLIT/PT.01.03/2024). The funders had no role in study design, data collection and analysis, decision to publish, or preparation of the manuscript.

### Grant Disclosures

The following grant information was disclosed by the authors:
Ministry of Education, Culture, Research, and Technology through a postgraduate research scholarship administered by the Directorate of Research, Universitas Gadjah Mada: 048/E5/PG.02.00.PL/2024; 2781/UN1/DITLIT/PT.01.03/2024.

### Competing Interests

The authors declare there are no competing interests.

### Author Contributions

- Cahyo Budiyanto conceived and designed the experiments, performed the experiments, analyzed the data, prepared figures and/or tables, and approved the final draft.
- Andriati Ningrum conceived and designed the experiments, authored or reviewed drafts of the article, and approved the final draft.
- Agnes Murdiati conceived and designed the experiments, authored or reviewed drafts of the article, and approved the final draft.
- Retno Indrati conceived and designed the experiments, analyzed the data, authored or reviewed drafts of the article, and approved the final draft.

### Animal Ethics

The following information was supplied relating to ethical approvals (i.e., approving body and any reference numbers):

Integrated Laboratory for Research and Testing, Universitas Gadjah Mada (no. 00073/04/LPPT/I/2024).

### Data Availability

The raw data are available in the Supplemental File.

## Supplemental Information

Supplemental information for this article can be found online at http://dx.doi.org/10.7717/peerj.19262#supplemental-information.

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

# PeerJ

of a prokaryotic homologue of the mammalian oligopeptide—proton symporters, PepT1 and PepT2. *The EMBO Journal* **30(2)**:417–426 DOI 10.1038/emboj.2010.309.

**Ohanenye IC, Sun X, Sarteshnizi RA, Udenigwe CC. 2021.** Germination alters the microstructure, *in vitro* protein digestibility, $\alpha$-glucosidase and dipeptidyl peptidase-IV inhibitory activities of bioaccessible fraction of pigeon pea (*Cajanus cajan*) seeds. *Legume Science* **3(1)**:e79 DOI 10.1002/leg3.79.

**Pak VV, Kim SH, Koo M, Lee N, Shakhidoyatov KM, Kwon DY. 2006.** Peptide design of a competitive inhibitor for HMG-CoA reductase based on statin structure. *Biopolymers* **84(6)**:585–594 DOI 10.1002/bip.20580.

**Palupi HT, Estiasih T, Yunianta, Sutrisno A. 2022.** Physicochemical and protein characterization of lima bean (*Phaseolus lunatus* L) seed. *Food Research* **6(1)**:168–177 DOI 10.26656/fr.2017.6(1).107.

**Patel G, Misra A. 2011.** 10 - oral delivery of proteins and peptides: concepts and applications. In: Misra A, ed. *Challenges in delivery of therapeutic genomics and proteomics.* London: Elservier Inc., 481–529 DOI 10.1016/B978-0-12-384964-9.00010-4.

**Pertiwi MGP, Marsono Y, Indrati R. 2020.** *In vitro* gastrointestinal simulation of tempe prepared from koro kratok (*Phaseolus lunatus* L.) as an angiotensin-converting enzyme inhibitor. *Journal of Food Science and Technology* **57**:1847–1855 DOI 10.1007/s13197-019-04219-1.

**Ratnayani K, Suter IK, Antara NS, Putra INK. 2019.** Angiotensin converting enzyme (ACE) inhibitory activity of peptide fraction of germinated pigeon pea (*Cajanus cajan* (L.) Millsp.). *Indonesian Journal of Chemistry* **19(4)**:900–906 DOI 10.22146/ijc.37513.

**Sandoval-Sicairos ES, Domínguez-Rodríguez M, Montoya-Rodríguez A, Milán-Noris AK, Reyes-Moreno C, Milán-Carrillo J. 2020.** Phytochemical compounds and antioxidant activity modified by germination and hydrolysis in mexican *Amaranth.* *Plant Foods for Human Nutrition* **75**:192–199 DOI 10.1007/s11130-020-00798-z.

**Schönfeldt H, Pretorius B, Hall N. 2016.** Bioavailability of nutrients. *Encyclopedia of Food and Health* **1**:401–406 DOI 10.1016/B978-0-12-384947-2.00068-4.

**Seidu KT, Osundahunsi OF, Osamudiamen PM. 2018.** Nutrients assessment of some lima bean varieties grown in southwest Nigeria. *International Food Research Journal* **25(2)**:848–853.

**Sepúlveda FV, Smith MW. 1978.** Discrimination between different entry mechanisms for neutral amino acids in rabbit ileal mucosa. *The Journal of Physiology* **282(1)**:129–151 DOI 10.1113/jphysiol.1978.sp012449.

**Sepúlveda FV, Smith MW. 1979.** Different mechanisms for neutral amino acid uptake by new-born pig colon. *The Journal of Physiology* **286(1)**:479–490 DOI 10.1113/jphysiol.1979.sp012632.

**Silva MBde-Ce, Souza CAda-C, Philadelpho BO, Cunha MMNda, Batista FPR, Silva JRda, Druzian JI, Castilho MS, Cilli EM, Ferreira ES. 2018.** *In vitro* and *in silico* studies of 3-hydroxy-3-methyl-glutaryl coenzyme A reductase inhibitory activity of the cowpea Gln-Asp-Phe peptide. *Food Chemistry* **259**:270–277 DOI 10.1016/j.foodchem.2018.03.132.

**Smith ME, Morton DG. 2010.** The digestive system. In: *Digestion and absorption*. 2nd edition. New York: Churchill Livingstone.

**Suzuki T. 2013.** Regulation of intestinal epithelial permeability by tight junctions. *Cellular and Molecular Life Sciences* **70**:631–659 DOI 10.1007/s00018-012-1070-x.

**Xiao Y, Wei Q, Du L, Guo Z, Li Y. 2024.** *In vitro* evaluation and *in situ* intestinal absorption characterisation of paeoniflorin nanoparticles in a rat model. *The Royal Society of Chemistry Advances* **14(31)**:22113–22122 DOI 10.1039/d4ra03419h.

**Xu Q, Hong H, Wu J, Yan X. 2019.** Bioavailability of bioactive peptides derived from food proteins across the intestinal epithelial membrane: a review. *Trends in Food Science and Technology* **86**:399–411 DOI 10.1016/j.tifs.2019.02.050.

**Yuan Y, Li C, Zheng Q, Wu J, Zhu K, Shen X, Cao J. 2019.** Effect of simulated gastrointestinal digestion *in vitro* on the antioxidant activity, molecular weight and microstructure of polysaccharides from a tropical sea cucumber (*Holothuria leucospilota*). *Food Hydrocolloids* **89**:735–741 DOI 10.1016/j.foodhyd.2018.11.040.