# Peer review of "A novel approach to assessing the bioavailability of biopeptide inhibitor of HMG CoA reductase from germinated and ungerminated Kara Kratok (*Phaseolus lunatus* L.)"

_PeerJ, doi:10.7717/peerj.19262_

## Round 0.1 · original submission · Major Revisions

Dear authors, I ask you to improve the manuscript very carefully. One of the main shortcomings is the lack of correct comparison of samples in Figures 2, 4, 5. Readers should see for which values ​​on the abscissa axis there are reliable differences between the samples, and for which they are absent (mark *, ** or ***). The methods you described (lines 169-172) are not used in the figures. Threefold replication is not used in serious studies. I recommend that you conduct several more experiments to achieve 6-7 replications. In this case, the data in the figures should be presented as a box analysis (median, first and third quartiles, minimum and maximum values). On the ordinate axis, it is advisable to round the values ​​to whole numbers (except for Figure 2B).

Reviewer 1 ·

Basic reporting

Dear colleagues, I am happy to be able to collaborate with your work. I congratulate you on your performance and dedication. I would like to ask for corrections and adjustments, which often go unnoticed when we are writing.

Write the scientific name in full once in the introduction, then use the abbreviated scientific name P. lunatus. It is preferable to use the scientific name of the species studied throughout the work; you can mention the name of the variety in the introduction once to help the reader understand the subject.

Only the scientific names of a species are written in italics. The names of varieties are not written in italics.

References should be standardized, with regard to the name of the journals and other information. It is preferable to write the name of the journals in full and in non-italic text, as well as the other data of the journals.

Line 82, the period after the name of the determiner is missing.

Lines 242, 243 and 248, remove the th from the minutes.

Lines 108, 280, 290, 339 and 346, correct the conjunction between the names of the authors. In citations, use (& or and) and do not alternate between the two.

In the caption of table 1, remove the second period or add other information if necessary.
Figures 1 and 2, add the period at the end of the sentence.
Figure 3: In line 20, where it reads ‘kara kratok(Phaseolus lunatus L.)’ it should read ‘kara kratok (Phaseolus lunatus L.)’. In line 22, the parentheses should be closed. In the same line where it reads ‘210, 240’ it should read ‘210 and 240’. In line 24 where it reads ‘dan’ it should read ‘and’.

Experimental design

In the introduction, the importance of the biopeptide inhibitor of HMG CoA Reductase should be highlighted. The species Phaseolus lunatus L. should also be explored, mainly addressing the theme of its importance for the population and the reason for choosing the species as the object of study.

Validity of the findings

no comment

Annotated reviews are not available for download in order to protect the identity of reviewers who chose to remain anonymous.

·

Basic reporting

Clearly outlines the significance of HMG CoA reductase inhibitors and the relevance of biopeptides derived from Phaseolus lunatus L. to this field in the introduction section.

Provide more background on previous studies related to bioavailability assessments of similar compounds

Consider discussing briefly the existing methods for assessing bioavailability in biopeptides, particularly focusing on any gaps your study aims to fill.

Experimental design

What was the rationale behind your chosen sample size for both germinated and ungerminated Kara Kratok?

Provide more details about the analytical techniques employed, and how do these techniques specifically contribute to validating your findings?

Validity of the findings

The discussion section should critically evaluate how your findings compare with existing literature on HMG CoA reductase inhibitors. Are there any discrepancies or unexpected results that warrant further exploration?

Have you addressed potential limitations in your study? Discussing factors such as variability in peptide extraction efficiency or differences in absorption rates could enhance the credibility of your conclusions.

Additional comments

Check for some grammatical and spelling errors to further improve the quality of this paper

·

Basic reporting

I have included my suggestions in the PDF manuscript. The manuscript looks good, is easy to read, and is understandable in most parts. Still, it needs some adjustments in the description of the results, corrections to some sentences in English, and attention to the use of present, future, and past tenses, as well as how to start certain phrases.

Experimental design

no comment

Validity of the findings

no comment

Additional comments

I have included my suggestions in the PDF manuscript. The manuscript looks good, is easy to read, and is understandable in most parts. Still, it needs some adjustments in the description of the results, corrections to some sentences in English, and attention to the use of present, future, and past tenses, as well as how to start certain phrases.

---

## Round 0.2 · accepted · Accept

Dear authors, I am pleased to inform you that your article has been accepted for publication.

·

Basic reporting

Previous comments addressed by the authors

Experimental design

Respondd already

Validity of the findings

Good and ready for publication

Additional comments

Consider for Acceptance